Morphometry of cellular behavior of coelomocytes from starfish Asterias amurensis

Karetin Yuriy A. yura15cbx@gmail.com
A.V. Zhirmunsky National Scientific Center of Marine Biology, Far Eastern Branch, Russian Academy of Sciences , Vladivostok , Russia
Riesgo-Escovar Juan
Electronic publication date: 2021 Nov 23
Publication date: 2021
Volume: 9
Electronic Location ID: e12514
Received 2018 Nov 13; Accepted 2021 Oct 27
Copyright: ©2021 Karetin
Copyright year: 2021
Copyright holder: Karetin
License: This is an open access article distributed under the terms of the Creative Commons Attribution License, which permits unrestricted use, distribution, reproduction and adaptation in any medium and for any purpose provided that it is properly attributed. For attribution, the original author(s), title, publication source (PeerJ) and either DOI or URL of the article must be cited.
License URL: https://creativecommons.org/licenses/by/4.0/

Keywords: Morphometry, Asterias Amurensis, Fractal analysis, Classification, Coelomocytes

Funding: The authors received no funding for this work.

==============================
A comprehensive statistical analysis using a wide range of linear and non-linear morphological parameters enabled identification of the main stages in the in vitro dynamics of cell behavior of immune cells of the marine invertebrate Asterias amurensis (Echinodermata, Asteroidea). Three stages may be distinguished in the cell behavior, which are characterized by the differences in complexity of the cell boundary microsculpture as well as by the size and asymmetry of the cell and convex hull of the cell. The first stage (5 min after placing cells onto a substrate) is characterized by more complex cell morphology and an increase in the process number and spreading area. The second stage (15 min) is characterized by simplification of cell morphology, retraction of some processes, and rounding of cells upon continued cell spreading. At the third stage (60 min), new large processes with rounded contours emerge due to partial retraction of the flattened cell surface. Each stage is characterized by statistically significant differences in several linear and nonlinear parameters of the external morphology for all cell types.

Introduction

Asterias amurensis is a seastar native to the seacoasts of China, Korea, Russia and Japan. This species recently has been introduced to the Tasmania, southern Australia, Alaska, the Aleutian Islands, parts of Europe, and Maine (Stevens, 2012). Today the species attracts the close attention of researchers in view of the wide spread, threatening the stability of biocenoses on the introduced territories (Ross, Johnson & Chad, 2002; Dommisse & Hough, 2004).

The typical parameters for classification of invertebrate immune cells include features such as the cytoplasm granularity, nuclear-cytoplasmic relationships, phagocytic activity and immunological specificity. Most of these features are variable (Fisher, 1986). For example, granulocytes may have the agranular precursors (Hine, 1999); according to Martin & Hose Jo (1992), granules can appear upon condition changes or be present in cells defined as hyalinocytes based on other morphological features. The ability of marine invertebrate immune cells to phagocytosis can significantly differ among animals within a single population (Kurtz, 2002); phagocytosis can be observed only in one type (Scapigliatia & Mazzinia, 1994) or in all described types of hemocytes (Stoepler et al., 2013). Invertebrates also may mount a highly variable immune response that is dependent on which pathogen is involved (Cerenius & Söderhäll, 2013). The antigens characterize distinct subsets which partially overlap with those defined by morphological description (Kurucz et al., 2007). In some species only individual cell types became immunologically activated with obvious morphological changes (Hwang et al., 2015).

Since the conventional parameters often provide an ambiguous picture, the development of a universal classification scheme requires a combination of more extensive morphometric, genetic and biochemical methods as well as cytodifferentiation data of determining the cell specificity. Different sets of parameters will potentially able to identify different cell types associated with a particular feature of their physiology or morphology. Which of classification models can serve as a natural universal model for the classification hemocytes it is still an open question. However, we may suppose that the external morphology play an important role in the natural classification of invertebrate immune cells. Features of morphology of the flattened cell shape are reflected in its cytoskeleton structure, the organization of related cell systems and directly dependent on the type-specific cell behavior.

The methodology of multiparametric morphological classification of immune cells of invertebrates, using nonlinear parameters of morphology analysis (Karetin, 2010a; Karetin, 2010b; Karetin & Pushchin, 2015) and interspecific comparisons of morphotypes of invertebrate cells based on such a methodology (Karetin, 2016) show their effectiveness in describing the morphology of hemocytes and coelomocytes of marine invertebrates. However, a description of static cell morphology, which is usually carried out in a few minutes after the attachments, makes classification schemes based on morphological features incomplete. Analysis of the dynamics of cellular behavior shows that we are dealing with a number of transitional forms, the dynamics of which can also reflect the species-specific features of cellular behavior, the state of the animal’s immune status or reaction to the experimental effect. Description of cell dynamics should complement the static patterns of cellular morphology used for description of the cellular immunity of invertebrates.

Using the population of Asterias amurensis coelomocytes as example, we have shown in this work the effectiveness of multiparametric analysis of the dynamics of the spreading of immune cells, with the identification of a number of characteristic morphotypes of cells and their transformation in the process of short-term cultivation.

Materials & Methods

Material preparation

The study was performed on 398 coelomocytes of the 10 Asterias amurensis starfish (Echinodermata, Asteroidea) (Lutken, 1871). Field experiments were approved by the Federal Fisheries Administration, Primorsky Territorial Administration, issued for NSCMB FEB RAS. Approval number to access field sites: 252021030802. Animals were collected in the Peter the Great Bay (Sea of Japan, Russia). After collecting, the coelomic fluid was sampled from the starfish coelomic cavity using a syringe, placed onto glass coverslips, and incubated for 2, 5, 15, and 60 min. After incubation the glasses with adhered cells were fixed with a 4% formaldehyde solution in seawater, stained with hematoxylin and eosin, dehydrated, and embedded in Canada balsam for a light microscopy study. Photos of the adhered cells were taken using a 10-megapixel digital camera of a Zeiss Axiovert 200M Apotome microscope (magnification of the ×100 lens). The images of the cells were drawn directly from a tablet computer screen onto transparent films that were then scanned. Both silhouette and contour images of flattened cells were analyzed. A single pixel outline of the outer cell boundary was used for contour images. Cell images were digitized and converted to single-bit format.

Parameterization

Parameters were obtained using ImageJ 1.41 image analysis software with a FracLac 2.5 plug-in for a fractal analysis. 1/2half in50/out50 was calculated using CellMaster 1.3.2b, https://figshare.com/s/d3c3784e43efa1913db4, author: Eduardas Cicinskas.

A total of 39 parameters were examined (see Supplemental 1); of these, 19 parameters were selected for use, including seven nonlinear and 12 linear parameters.

Linear parameters:

Aspect ratio: major_axis/minor_axis of cell image, (AR);

Roundness: 4*area /(π*major_axis2), (Round);

1/2half: the ratio of area parts of the cell in the two halves of the bounding circle. The diameter dividing cells was drawn in the direction that most unevenly divided the image of cells. Parameter was used as a measure of cell asymmetry;

in50/out50: The ratio of cell area in the outer half of the bounding circle to the area of the cells in the inner half of bounding circle;

Area;

Circularity: 4 π*area / perimeter2, (Circ);

Feret’s Diameter: the longest distance between any two points along the object boundary, also known as maximum caliper, (Feret);

Hull’s Perimeter: perimeter of the convex hull drawn around the object. The convex hull is a boundary enclosing the foreground pixels of an image using straight line segments to each outermost point, (Hull’sPer);

Diameter of Bounding Circle: the smallest circle around the convex hull, (DiamBoundCirc);

Density: Foreground Pixels/Hull Area;

Perimeter of cell, (Per);

Hull’s Perimeter: perimeter of the convex hull drawn around the object (Hull’sPer);

Nonlinear parameters:

Lacunarity L (F - foreground pixels): lacunarity based on the variation of pixels in each analyzed box summarized over all grid orientations for an image, only pixels of the image of the object were taken into account, (LF);

Lacunarity L (E - foreground and empty space): lacunarity calculated on the basis of differences in the number of pixels in each square of the grid for all orientations of the squares, with allowance for the background pixels taken as zero, (LE);

outMeanΛD: The lacunarity of contour images of cells based box counting dimension (ΛD B) or its average ΛD B = (Grids ∑G =1(ΛD B(G))) ×Grids−1 , where ΛD B(G) is the lacunarity calculated for each orientation of the grid, Grids are all possible orientations of the grid;

MeanΛ: Average lacunarity calculated for different orientations of the square grid;

LCFD Prefactor Lacunarity: a measure of heterogeneity or translational invariance dependent on where a grid series is placed, Local Connected Fractal Dimension is found from a type of fractal analysis that uses pixel mass from concentrically placed sampling units, using the connected set around each pixel to produce a distribution of local variation in complexity, (LCFD PreLac);

Fractal dimension of contour images of cells calculated as follows: Mean D = Σ(D)/GRIDS (D =slope (ln(Boxes with Foreground Pixels)/ln(ɛ)), where ɛ - box size or scale, the average D B (box-counting fractal dimension) from multiple box counting scans, each delivering its own D B, based on a different orientation of grid, (outMeanD);

Mean Local Fractal Dimension: the average dimension calculated on the basis of multiple sampling an image and defining the local fractal dimension of various parts of contour images of cells, (outMeanLFD).

Statistical analysis

All analyses were performed using NCSS 2007 and STATISTICA 10 software.

The correlation between parameters was measured using Pearson’s linear correlation analysis. The scatterplots of pair-wise correlations were also examined to reveal false correlations due to outliers (Gordon, 1999). In each group of significantly correlated parameters (R ≥ 0.8 at p < 0.05), only one parameter was selected to assure that the respective aspect of cell morphology is adequately, but non-redundantly, represented (Schweitzer & Renehan, 1997).

Cells were classified using a hierarchical cluster analysis; the Ward’s agglomerative algorithm providing the best results in classifying cell objects on the basis of morphological parameters was used as a clustering method (Pushchin & Karetin, 2009). It starts with each object as a separate cluster, merging similar objects into successively larger clusters. Analysis of variance is used at each cycle to minimize the within-cluster variance.

Tukey-Kramer Multiple-Comparison Test is a single-step multiple comparison procedure that was used to find means of groups that are significantly different from each other. It compares all pairs of means, and is uses a studentized range distribution (q).

Results

Visual description of cell morphology

Two minutes after placing cells onto glass (group 1), cells that could spread and form a large number of small and large processes were observed among a large number of small cells with minor processes. Cells fixed 5 min after placement (group 2) were the most heterogeneous group in the experiment. The group included both small cells with minor processes and cells of a very complex structure, with long processes covered by secondary microprocesses, branched processes, and processes with expanding ends. After 15 min of culturing (group 3), cells remained visually very flattened, but acquired in most cases a more regular nonsegmented shape compared to cells of the previous group. Often, their shape was nearly round; the cells had a small number of thin and long processes or a large number of small processes evenly covering the outer contour of the cell. After 60 min of culturing (group 4), the number of small cellular processes was visually even more reduced; the processes lost sharpness; the microsculpture of cell boundaries became simpler; rounded cell forms at this stage alternated with cells having a small number of large processes (Fig. 1). At this stage, a morphotype emerged that had been absent at the earlier cultivation times: cells with curved, circle segment-like processes. Therefore, starting with the 15th min of culturing, simplification of the cell morphology and retraction of small processes occurred along with continued cell spreading. At the 60th min of culturing, retraction led to the disappearance of certain cell regions: there remained elements of the region boundary in the form of curved pseudopodia delineating the boundary.

Figure 1 Silhouette images of cells at different stages of cultivation: (A) 2 min (group 1); (B) 5 min (group 2); (C) 15 min (group 3); (D) 60 min (group 4).

Trends in parameter dynamics

The dynamics of cultured cell parameters demonstrated 5 trends: 1. an increase or drop in the parameter value during culturing, with a significant difference of each group (ex. Area) (Fig. 2A); 2. a sharp decrease or increase in the studied parameter value in group 2 (ex. Circularity) (Fig. 2B); 3. a significant increase in the parameter value in the period from the 2nd to 5th min and from 15th to 60th min of culturing, with an insignificant difference between groups 2 and 3 (ex. Feret’s diameter) (Fig. 2C); 4. an increase or decrease in the parameter value, starting with group 3, with an insignificant difference between groups 1 and 2 (ex. outMeanLCFD) (Fig. 2D); 5. significantly smaller parameter values in groups 1 and 3 and significantly larger parameter values in groups 2 and 4 (ex. in50/out50) (Fig. 2E). A wide range of parameters had no significant differences among all four groups.

Figure 2 Different trends in changes in parameter values during in vitro culturing of coelomocytes from A. amurensis.

(A) Area; (B) circularity; (C) Feret’s diameter; (D) mean local connected fractal dimension of contour images of cells; (E) in50/out50.

Parameters showing similar trends were usually highly correlated, although there were exceptions from this rule. Parameters, changes in which during culturing had no recognizable trends, were not of great interest for this study because they did not reflect the dynamics of morphological transformations of cultured cells; although, different cell types within a single time group showed significant differences in some parameters (AR, Round), and some of these parameters (e.g., 1/2half) became key parameters for classifying cells within the same time group in other species of marine invertebrates (Karetin & Pushchin, 2015). In general, according to the Tukey-Kramer multiple comparison test, cells of all 4 time groups had significant differences in a number of studied parameters (Table 1).

Table 1 Tukey-Kramer multiple comparison test of differences in five parameters in four time groups.

	Group	Mean	Different from groups	
Area				
	2 min	45644.3 5	(P = 0,02401), 15 (P = 0,00000), 60 (P = 0,00000)	
	5 min	63175.76 2	(P = 0,02401), 15 (P = 0,00000), 60 (P = 0,00000)	
	15 min	88625.09 2	(P = 0,00000), 5 (P = 0,00000), 60 (P = 0,00000)	
	60 min	151098.5 2	(P = 0,00000), 5 (P = 0,00000), 15 (P = 0,00000)	
Circularity				
	5 min	0.1707483 15	(P = 0,00000), 2 (P = 0,00000), 60 (P = 0,00000)	
	15 min	0.2614072	5 (P = 0,00000), 60 (P = 0,00873)	
	2 min	0.290832	5 (P = 0,00000)	
	60 min	0.2964816	5 (P = 0,00000), 15 (P = 0,00873)	
Feret’s diameter				
	2 min	1.078258	5 (P = 0,0000), 15 (P = 0,0000), 60 (P = 0,0000)	
	5 min	1.520865	2 (P = 0,0000), 60 (P = 0,0000)	
	15 min	1.584021	2 (P = 0,0000), 60 (P = 0,0000)	
	60 min	2.144441	2 (P = 0,0000), 5 (P = 0,0000), 15 (P = 0,0000)	
in50_out50				
	15 min	1.328015	60 (P = 0,00003), 5 (P = 0,00000)	
	2 min	1.486176	60 (P = 0,04257), 5 (P = 0,00000)	
	60 min	1.797293	15 (P = 0,00003), 2 (P = 0,04257), 5 (P = 0,00082)	
	5 min	2.222238	15 (P = 0,00000), 2 (P = 0,00000), 60 (P = 0,00082)	
LE				
	60 min	1.709738	15 (P = 0,00000), 5 (P = 0,00000), 2 (P = 0,00000)	
	15 min	2.075289	60 (P = 0,00000), 5 (P = 0,00025), 2 (P = 0,00000)	
	5 min	2.296859	60 (P = 0,00000), 15 (P = 0,00025), 2 (P = 0,01327)	
	2 min	2.473771	60 (P = 0,00000), 15 (P = 0,00000), 5 (P = 0,01327)	

Peculiarities of parameter selection

A parameter selection methodology used in our previous studies (Pushchin & Karetin, 2014; Karetin & Pushchin, 2015) involved the initial search for highly correlated characteristics and selection of those that had a higher multimodality index value. In the present study, the methodology needed modifying because characteristics highly correlated in the same time group were not correlated in another, and many parameters with similar trends in parameter value changes had almost identical multimodality indices that differed by no more than 0.01. Parameters with a lower multimodality index were excluded from the parameters highly correlated in all 4 time groups as well as from the parameters with similar trends. In the case of similar multimodality indices, we chose a parameter providing a better cluster structure of identified cell types; highly correlated parameters with a similar trend in the parameter dynamics as well as parameters without significant differences in all time groups were excluded from the cluster analysis and description of intergroup differences.

The lack of correlation for similar trends and vice versa as well as the difference in correlation in different time groups generally indicate that this parameter in different cell groups describes different features of cell morphology.

Characteristics of cell morphology described by the parameters used

A complex of linear and quasi-fractal parameters was used to parametrize the cells. Linear parameters include morphometric parameters based on Euclidean geometry, such as the area, the perimeter of the cell, a number of parameters of the bounding circle and the convex hull. The area reflects the size of the cell and, to a greater extent, the spreading area, which can be many times larger than the original size of the cell, the perimeter depends on both the size of the cell and the number of processes, the bounding circle and the convex hull (a boundary enclosing the foreground pixels of an image using straight line segments to each outermost point, Fig. 3) reflect the area of cell stratification defined by the longest processes. Also, the linear parameters include density, which reflects the ratio of the number of pixels in the image to the total number of pixels covered by the circle bounding the cell. The value of density depends on the ratio of the size of the cell body to the area of stratification of its processes.

Figure 3 Convex hull of cell.

The parameters Roundness and Circularity characterize various aspects of the cell shape, Roundness characterizes the deviation of the shape of the object from the circle towards the ellipse; Circularity, determined by the ratio of the area to the perimeter, is equal to one for the circle and decreases as the perimeter of the cell border increases, that is, it depends not only on the overall shape, as in the case of Roundness, but also on the number of cell processes that increase the perimeter of the cell.

The parameter 1/2half is the ratio of area parts of the cell that turned out to be in the two halves of the bounding circle most unevenly divided the image (in Fig. 4: 1/2half = A/B). The largest values of this parameter will belong to the cells most asymmetrically positioned inside the bounding circle. The in50/out50 parameter describes the ratio of the cell area in the inner and outer parts of the bounding circle (in Fig. 4: in50/out50 = C/D), the parameter value is determined both by the asymmetry of the cell location inside the bounding circle, and by the ratio of the cell body size and the area of stratification of the cell processes.

Figure 4 1/2half and in50/out50.

calculation. 1/2half = A/B , where A, B –two segments of the cell lying in different halves of the bounding circle . in50/out50 = C/D, where C and D are the parts of the cell located in the inner and outer part of the.

Nonlinear parameters include a number of fractal dimensions and lacunarities, the value of which depends on the complexity of the object or the heterogeneity of its space filling. Fractal analysis is based on the consideration of an object as a quasi-fractal, which has a fractional dimension, the values that describe them are the same as objects of Euclidean geometry describes by measures of length, area, or volume. The basic algorithm for calculating the fractal dimension used here is the box-counting method, for counting which an object is enclosed in a network of squares of a certain size (Fig. 5). The number of squares that the structure fell into is calculated, and the ratio of the decimal logarithm of the number of squares that covered the structure to the decimal logarithm of the length of the side of the square is determined. In the next step, the structure is covered by a network of smaller squares. Again, we calculate the ratio of the logarithms of the number of squares and the length of their sides. We do this procedure several times. The resulting dimension value is calculated using the formula: (1) D= limL→0lnNLlnL

Figure 5 Box-counting fractal dimension calculation.

See the description of the method in the text.

where N (L) is the number of squares with side L required to cover the fractal structure. Different types of fractal dimensions have a similar basic algorithm and can differ in such aspects as the shape of the elements in which the object is enclosed (it may not be a square grid, but a set of circles of increasing diameter, mass–radius dimension), the location of the grid elements: fixed squares or a rotating grid (MeanD), the calculation can be performed for the entire image as a whole (MeanD), or the local dimension in the vicinity of each pixel (Local fractal dimension) is calculated and averaged. To calculate the dimension of only the upper and lower scale of the image, only a large (BiggestD) or small (SmallestD) grid is used, respectively. The fractal dimension in our work was measured in contour and silhouette images. Silhouette images represent a planar one-bit image of a flattened cell, in the contour image only the one-pixel contour of the silhouette image of the cell remains. The fractal dimension of silhouette and contour images is significantly different, since the value of the fractal dimension of the silhouette image increases by the space filled with image pixels inside the cell, while the fractal dimension of contour images consists only of the fractalization of the cell boundaries.

Twofold increase the complexity of the cell shape leads to an increase in its fractal dimension by about 0.1 (Jelinek and Fernandez, 1998). Different types of fractal dimensions describe different aspects of the complex shape of a flattened cell. These differences are difficult to formalize in traditional terms of Euclidean geometry or classical descriptions of cell morphology, but they reliably differentiate different cell types.

Cell types

Because the visual analysis revealed that cells of each time groups were a morphologically heterogeneous population, cells of all groups were divided into morphotypes using a hierarchical cluster analysis. Five weakly correlated parameters with different trends in the parameter dynamics during culturing were selected for the cluster analysis: Area, Circ, Feret, in50/out50, and LE. The parameters were used to generate a cluster model for each time group, which clearly revealed three clusters corresponding to three cell morphotypes (Fig. 6). Four clusters might be distinguished in a cell sample from group 3; however, 3 clusters were also chosen in this group to simplify intergroup comparisons. This is consistent with the logic of hierarchical cluster analysis when closely related types can always be combined into a common parent type; cells of the parent type combine characteristics of subsidiary cell types and, to the same or greater extent, differ from cells of other clusters, like cells of subsidiary clusters.

Figure 6 Cluster diagrams for 4 time groups, plotted using Area, Circ, Feret, in50/out50, and LE parameters (A) 2 min; (B) 5 min; (C) 15 min; (D) 60 min. Ward‘s method, Euclidean distance measure.

Cells of obtained types differed in a number of the analyzed parameters. In this case, similarity of trends in parameter changes among types in each time group suggested that the cluster analysis identified a similar cell types in each group by transposing types 1 and 2 in the second group, which did not change the general picture because a cluster sequence may be arbitrary due to free rotation of cluster axes, not affecting results of the cluster analysis. In general, the identified cell types are characterized by different sizes and complexity of cell boundaries. Here, we may distinguish small cells with simple boundary microsculpture: 2-1, 5-2, 15-1, and 60-1 (type I); medium-sized cells: 2-3, 5-3, 15-3, and 60-3 (type II); large cells or cells with very complex dendritic microsculpture of boundaries: 2-2, 5-1, 15-2, and 60-2 (type III) (Fig. 7).

Figure 7 Silhouette images of type I–III cells from all time groups.

The parameter Area most clearly distinguished both time groups and individual cell types within time groups (Fig. 8A). It is seen that cell types include the smallest (type I), medium (type II), and large (type III) cells. In this case, the area of cells of similar types in different time groups increases within the 2 to 60 min culturing interval. Because we consider a single population of cells, this indicates that the spreading process that is accompanied by an increase in the area of the two-dimensional cell projection occurs in all cell types throughout the cultivation period.

Figure 8 Plots of means and 95% confidence intervals for parameters of identified cell types: (A) area, (B) Feret’s diameter, (C) perimeter, (D) Hull’s perimeter, (E) roundness.

When considering trends of individual cell types, parameters that are correlated with Feret’s Diameter, characterized by trend 3, and have an insignificant difference between time groups 2 and 3 demonstrate a continuous increase in the parameter value, with significant differences among all cell types within each group; however, group 3 has a lower upper boundary for feature manifestation compared to that of group 2, with the lower boundary being higher, which makes group 3 comparable to group 2 in the mean feature value (Fig. 8B). The Feret’s diameter is a dimensional characteristic correlated with parameters of the convex hull and bound circle, such as Hull’sPer and DiamBoundCirc. Both the total cell area and processes size affect values of all these parameters. An increase in the total cell area, which is well detectable in small type I cells with minor processes, is compensated by a decrease in complexity of cell boundaries in type II and III cells of group 3, which creates the effect of absent significant differences in these parameters between groups 2 and 3.

A group of parameters demonstrating a sharp drop or increase in a studied parameter value in group 2 (trend 2), which includes parameters Circularity, Density, and LF, demonstrates a small increase or decrease in parameter values upon transition from the Ist to IIIrd cell type. In this case, confidence intervals of values in most cases overlap both among cell types of the same group and among different time groups. The exceptions are cell types 5-3 and 5-1 that sharply distinguish group 2 from other groups. Decreased circularity and cell image density and increased lacunarity always non-specifically characterize an increase in structural complexity of the cell, cell protrusivity, complexity of boundaries, elongation, and asymmetry of filling of the surrounding space by the cell. The same pattern is observed for parameters associated with trend 5 (Perimeter, in50/out50): upon similar parameter dynamics for most cell types, types 5-1 and 6-2 are characterized by a sharp increase in values of the parameters Perimeter and in50/out50, which also indicates increased protrusivity in these cell types (Fig. 8C).

In general, it is noteworthy that the cell type 5-1 significantly surpasses the other cell types in most parameters that directly (several nonlinear parameters: outMeanLD, LF, MeanL, LCFD PreLac) or indirectly (several linear parameters: Circ, Density, in50/out50, Per) characterize spatial complexity of the cell. It is the type that includes cells reaching the highest level of spatial complexity at the 5th minute.

Parameters based on measurements of the convex hull and bound circle (Hull’sPer, DiamBoundCirc) (Fig. 8D) combine characteristics of cell size and the degree of protrusivity and are reducible to characteristics of filling of the surrounding space by the cell. Two cell types are clearly distinguished by high values of these parameters: 5-1–cells with the highest degree of protrusivity and 60-2–cells with the largest area and protrusivity that reduces to the 15th min of culturing and re-emerges at the 60th min, which is associated with partial retraction of the flattened cell body leaving large curved processes, as described above.

In addition, it can be seen that cell types 15-2 and 60-2 are significantly different in parameters Hull’s Circularity and Roundness (Fig. 8E); the other cell types do not have significant differences in these parameters. The Roundness describes similarity of the overall cell shape to the circle (4*area/(π*major_axis of fitted ellipse2), i.e., the highest Roundness value is observed for cells with an elongated shape typical of crawling cells or cells with long asymmetrically arranged processes. Large errors of the parameter’s mean and the absence of significant differences among most of the cell types indicate that a chaotic variety of common cellular shapes does not reflect any structural and functional patterns occurring in cell culture by the 15th minute of culturing. Only a significant increase in this parameter in types 15-2 and 60-2 at the 15th and 60th minutes of culturing means rounding of cells, which indicates the disappearance of large processes and a reduction in cell motility that leads to the development of radial symmetry. Also, rounding of cells is reflected in an increase in the Hull’s Circularity parameter value. The Circularity parameter that is based on the ratio of the object perimeter to the object area (4 π*area/perimeter2) reduces as the perimeter increases, i.e., it is anti-correlated with many nonlinear parameters that increase as indentation of cell boundaries becomes more pronounced, which is confirmed by the correlation analysis. However, Circularity applied to the convex hull connecting only extreme points of an object becomes a parameter that characterizes the overall shape of the object.

Correlation analysis

Interestingly, the presence of significant correlations for some pairs of parameters only in the 2nd time group is related to the prevalence of cells with much more complex morphology (type 5-1) in this group compared to other groups. For example, two parameters: Area − outMeanD (0.01, 0.70, 0.05, and 0.18; correlations are provided for groups 1 −4, respectively; significant and high correlations are shown in bold) demonstrate a significantly high correlation in the second group because a simultaneous significant increase in the area and spatial complexity of the cell occurs only in this group. A variety of both large and small cells with simple microsculpture (and, correspondingly, with a lower outMeanD value) makes the correlation between these parameters in other groups insignificant. We observed the same effect of spatial complexity of the cell contour on correlation of characteristics in a pair of characteristics: Feret − Density (−0.14, −0.80, −0.30, −0.58). An increase in Feret, a parameter describing one of the characteristics of space filling by the cell, is anti-correlated with Density only if an increase in the filling area is achieved through an increase in the length and number of cellular processes (and, correspondingly, through a decrease in the ratio of the number of object pixels to the number of background pixels), but not through continuous filling of the space by the cell body; otherwise, these two parameters are positively correlated. In other groups, an increase in the Feret value is achieved both through an increase in the size of flattened cells and through an increase in the number of processes, which makes the correlation invalid.

Some parameters have sufficient discriminatory power for reliable identification of the most dendritic cells combined in types 5-1 and 60-2. These parameters also form variable correlations in different time groups. One of these pairs is Density − Perimeter (0.31, −0.76, −0.42, −0.70). In groups 2 and 4, low density of the cell image in dendritic cells with a high degree of cell boundary indentation is anti-correlated with a large length of the cell perimeter; in groups 1 and 3, whose cells have less complex morphology, the density and perimeter grow simultaneously as the cell size increases, due to continuous filling of the surrounding space by the cell body, which makes the correlation insignificant.

In some cases, the difference in correlation in different time groups is associated with different dynamics of parameter changes in cells with different morphology. For example, in a pair of parameters outMeanD − outMeanLFD (0.73, 0.21, 0.66, 0.34), only the parameter outMeanD grows as protrusivity strongly increases in cells of groups 2 and 4; outMeanLFD is not correlated with the maximum increase in spatial complexity of the cell; the greatest outMeanLFD value occurs in a special type of cells with a relatively uniform distribution of processes around the cell body (Fig. 9). The morphology of these cells can be called moderately complex, and outMeanD values of these cells are medium. In groups 1 and 3, where this cell type reaches the maximum complexity of cell morphology, outMeanLFD is correlated with outMeanD; in groups with a sufficient number of cells with higher spatial complexity, a further increase in the outMeanD parameter is not accompanied by an increase in the outMeanLFD parameter, and the correlation is not valid.

Discussion

Coelomic fluid cells in invertebrates perform the protective function that is realized through several variants of a single systemic cellular response to a foreign body: phagocytosis of microscopic foreign particles identified by the cell, encapsulation − in vivo cell spreading across a foreign body that is too big to be uptaken by the cell, and, finally, cell spreading through a substrate as an attempt to isolate the foreign surface, the amount of which significantly exceeds the cell size (Ratner & Vinson, 1983).

Within a few minutes after placing coelomic fluid onto an artificial substrate, cells precipitate, adhere to it, form conglomerates of various sizes (a reaction of thrombus formation and cell agglutination of a body fluid, which is transferred to in vitro condition) and flatten. In this case, some species form almost a continuous cell layer, while others form a network of contiguous pseudopodia of cells self-assembled into cell flows.

Figure 9 Cell morphology corresponding to the different values of outMeanD and outMeanLFD parameters.

Morphological characteristics of coelomocytes vary greatly in different species of echinoderms, even within the same class. In the late 1950s, R.A. Boolootain and A.C. Guise from the Stanford University, on the basis of previously conducted studies of morphological features, described in detail various cell types of coelomocytes in 15 species from different classes of echinoderms (Boolootain & Guise, 1958). The developed classification of formed elements of the coelomic fluid amounted to at least 13 cell types. According to the authors, different cell morphotypes in some cases reflected the functional state of cells of the same types. The described cell elements are observed in all classes of echinoderms, except sea stars, in which amoebocytes are considered to be the only cellular component of the immune system (Bossche Vanden & Jangoux, 1976).

Introduction of electron microscopy, histochemical, and histological methods has enabled identification of five main cell types, the existence of which is now recognized by most researchers: granular and agranular amoebocytes, morula cells, and lymphocyte-like and flagellated cells (Korenbaum, 1989; Kudryavtsev, 2006). However, various names of the same classes of starfish coelomocytes can still be found in the literature. Furthermore, genesis of different cell types still remains unclear: whether they belong to several differentiation lines, or they stem from common progenitor cells. The amount of different coelomocyte types in the coelomic fluid depends on a wide range of external and internal factors, including different techniques for coelomic fluid fixation, which may affect the formula of cellular composition (Bossche Vanden & Jangoux, 1976).

The classification of adherent cells is even more ambiguous. It has been supposed that small cells are progenitors of all other cell types and that irreversible transformation occurs upon transition from small cells to differentiated cells, and from circulating cells with short pseudopodia to flattened cells (Fontaine & Lambert, 1975; Bossche Vanden & Jangoux, 1976). According to Pinsino, Thorndyke & Matranga (2007), adherent Asteroidea immune cells form cell shapes described as petaloid or filopodial cytoplasmic protrusions, which replace each other through a series of transitional forms for 8 min. In our study, cells with a small number of processes, which are especially numerous at the 15th min of culturing, as well as insufficiently flattened cells of a 2-minute culture may be assigned to the petaloid form. We describe for the first time the cell shape with curved processes, which emerges after 60 min of culturing.

Each time group contains cells with petaloid and filopodial cytoplasmic protrusions. The classification based on the most variable parameters distinguishing cell types with a high degree of reliability enabled us to identify types of small and petaloid cells (type I) that represent the initial and final stages of spreading across a cell substrate. These cells have the lowest, compared to other types, area, a simple form and boundary microsculpture, and respectively, a low value of the mean fractal dimension of the cell contour (outMeanD) and high Circularity. Type II is a transitional type between the petaloid and filopodial forms, which is characterized by medium values of most parameters. Type III includes predominantly cells of a filopodial form, which may be called the most morphologically differentiated cell form. These cells have high values of several fractal dimensions and lacunarities characterizing overall complexity of the cell shape and inhomogeneity of space filling by the form, a large spreading area, and even a greater spreading area of processes (low Density value), and a low Circularity value indicating complexity of the overall shape of this cell type, which is the most different from the rounded shape.

The identified cell types are comparable in each time group, but similar types differ in a number of characteristics in different time groups, and comparison of these differences enables identification of several trends in the dynamics of cell behavior. First of all, the degree of cell spreading continues to significantly increase in all cell types during culturing for 1 h. In this case, the maximum activity of cells was observed at the fifth minute of culturing. Here, there is a cell type (type 5-3) that significantly surpasses other types in microsculpture complexity and overall complexity of the cell shape. This indicates cell motility, active process formation, and cell shape changes. At this stage, cells are characterized by the search behavior; cell spreading also occurs most rapidly. By the 15th min, activity of cells reduces. The number of processes reduces; cells acquire a more rounded shape; the cell body area to protrusion area ratio increases; the number of petaloid type cells increases. Despite continued cell spreading, it looks, according to the patterns of morphological characteristics, like uniform spreading of a fixed cell with pronounced retraction of processes and uniform filling of a substrate by the cell body. This is confirmed by an increase in the cell area with a simultaneous reduction in most parameters that characterize complexity of cell morphology. Morphological complexity of cells at the 60th min of culturing decreases to a level comparable to that at the 2 min stage, but cells are characterized by a significantly greater degree of spreading and symmetry. Curved processes, which develop in this time group, make the cell shape more round, but do not reduce the Roundness parameter, which distinguishes these cells from typical filopodial cells at other spreading stages where increased protrusivity leads to a decrease in the Roundness value.

Conclusions

Unlike nerve tissue cells, the morphometry of which, both neurons (Pushchin & Karetin, 2014), and microglia cells (Morrison et al., 2017; Young & Morrison, 2018), is well developed, and uses a wide range of nonlinear parameters in combination with different imaging methods, the morphological description of fibroblast-like cells is less formalized. A large number of recent studies on biochemical, morphological, physiological, and genetic processes associated with changes in the immune cell behavior have used a statistically unverifiable language to describe cell shapes. The difficulty is related to irregularity and a wide chaotic variety of shapes of flattened cells. Attempt to cover the entire potential spectrum of cell shapes in vitro using classical morphometry inevitably results in a set of rather general, non-specific, linear morphometric parameters describing the general features of a two-dimensional object, such as the area, diameter of the bounding circle, circularity, or elongation of the cell (Rajagopalan et al., 2004; Solon et al., 2007). However, these structures can be effectively described by a set of nonlinear morphometric parameters, including calculation of the fractal and information dimension, evaluation of spatial heterogeneity and lacunarity of the object, its multifractal spectrum, etc. A number of examples involving a wide range of cell types have demonstrated the possibility, in principle, of efficient, comparative, numerical morphometry of cells (Karetin, 2010a; Karetin, 2010b; Karetin, 2016). This argues that the methodology based on the synthesis of linear and nonlinear morphometry can effectively complement a non-formalized language for description of morphology of invertebrates’ immune cells in culture.

A complex statistical analysis of the morphology of coelomocytes from Asterias amurensis (Echinodermata, Asteroidea), which included cluster and correlation analyses and an variance analysis using a variety of linear and nonlinear cell morphology parameters, enabled identification of cell characteristics, such as complexity of cell boundary microsculpture, overall asymmetry of the cell, sizes of the cell, bounding circle, and convex hull, and inhomogeneity of surrounding space filling by the cell and establishment of a relationship between these characteristics and differences in the dynamics of cell spreading at different culturing stages.

Supplemental Information

Supplemental Information 1 Complete list of parameters

Click here for additional data file.

Supplemental Information 2 Asterias amurensis hemocyte morphology dataset

Statistical analysis, graphs and the entire dataset of parameters of Asterias amurensis hemocyte morphology.

Click here for additional data file.

Additional Information and Declarations

Competing Interests

Author Contributions

Field Study Permissions

Data Availability

The authors declare there are no competing interests.

Yuriy A. Karetin conceived and designed the experiments, performed the experiments, analyzed the data, prepared figures and/or tables, authored or reviewed drafts of the paper, and approved the final draft.

The following information was supplied relating to field study approvals (i.e., approving body and any reference numbers):

Field experiments were approved by the Federal Fisheries Administration, Primorsky Territorial Administration, issued for NSCMB FEB RAS.

A copy of the document is attached in the Supplemental Files.

The following information was supplied regarding data availability:

Statistical analysis, graphs, and the entire set of parameter values are available in the Supplementary File.

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
