# Peer review of "Morphometry of cellular behavior of coelomocytes from starfish Asterias amurensis"

_PeerJ, doi:10.7717/peerj.12514_

## Round 0.1 · original submission · Major Revisions

Please heed the comments of the reviewers in submitting a new version of the paper, and please review the English.

Reviewer 1 ·

Basic reporting

See below

Experimental design

See below

Validity of the findings

See below

Additional comments

This is one interesting study which revels that in contrast to the morphology of neurons, classical morphometry of which is well developed, the morphological description of fibroblast-like cells is less formalized. The authors make and specially emphasis about the conventional cytological parameters often provide an ambiguous picture, the development of an universal classification scheme requires a combination of more complex morphometry as well as genetic and biochemical methods of determining the cell specificity and cytodifferentiation data.

In this work the authors used 400 coelomocytes of the Asterias amurensis as example, in order to show the effectiveness of multiparametric analysis of the dynamics of the spreading of immune cells, including the identification of a number of characteristic morphotypes of cells and their transformation in the process of short-term cultivation.

Could be interesting mention the amazing size increases across the time.

These cells were adequate prepared for a qualitative light microscopy study, and morphometrically analyzed, for this study, photography’s were take using a 10-megapixel digital camera of a Zeiss Axiovert 200M Apotome microscope.

Also all the 39 measurements from all cellular parameters were made adequate as well as their statistical analyses. Thus, linear and fractal analyses were calculated with the software (Image J) and 12 linear parameters were exhaustive analyzed in 11 measurements and non-linear in 7.

They found lack of correlation for similar trends and vice versa as well as the difference in correlation in different time groups. Thus, these founding’s generally indicate that this parameter in different cell groups describes different features of cell morphology.

In relation with the cell types, the parameters were used to generate a cluster model for each one, which clearly revealed three clusters corresponding to three cell morphotypes.

But, which one is the size of them?

They also, found that some parameters have sufficient discriminatory power for reliable identification of the most dendritic cells combined, and form variable correlations in different time groups. Also, included cluster and correlation analyses and an variance analysis using a variety of linear and nonlinear cell morphology parameters, enabled identification of cell characteristics, such as complexity of cell boundary microsculpture, overall asymmetry of the cell, sizes of the cell, bounding circle, and convex hull, and inhomogeneity of surrounding space filling by the cell and establishment of a relationship between these characteristics and differences in the dynamics of cell spreading at different culturing stages.

My principal comments are especially in the representation of the figures: Figures 1, 2, 4, why to put lettering below each graph and using point? (A. B. C. D. E.) also the codes will be better in the figure caption.

Figures 3, and 5. Although, the size of the type cells is clear, will be better to put an scale bar showing the micrometers.

In Table 1, it do not show the numbers of significant differences between groups.

·

Basic reporting

no comment

Experimental design

no comment

Validity of the findings

no comment

Additional comments

This paper constitutes an original study about morphometry of Asterias amurensis immune cells after collection, using a wide range of linear and non-linear morphological parameters including size, asymmetry, complexity, form, area, circularity, diameter etc. Despite the interesting focus of the paper I believe that the authors should make some changes and improvements of the text in order to make the paper suitable for the journal. I am most dubious about the parameters reported in the text and the presentation form choosen to report some data (see detailed comments). Also The English language should be improved to make comprehension less difficult and it is assumed a lot of sectoral knowledge for the reader and the readership. Considering this issue my suggestion to the authors is to revise and resubmit the paper in a new form.
Detailed comments:
Abstract:
- Line 49: what does “cell boundary microsculpture” stand for? Please define it
- Line 50: what does “convex hull of the cell” stand for? Please define it
Introduction
- Line 85-91: This part is not well-written. Please revise
MATERIALS AND METHODS
- Line 112: How many animals were used as donor?
- Line 114-118: This part is not well-written. Please revise
- Line 130-164: This part is not well-written and it assumes a lot of knowledge from the reader and the readership of the journal probably does not know what the major part of parameters selected are. This coud be helped with using the same style to explain each parameters: for example Aspect ratio (AR):brief explanation; Roundness (Round): brief explanation; and so on…..
“the ratio of area parts of the cell that turned out to be in the two halves of the bounding circle, the diameter dividing cells was drawn in the direction that most unevenly divided the image cells and was used as a measure of cell asymmetry, (1/2half): this part is really confusing and impossible to understand. The same “The ratio of cell
area in the outer half of the bounding circle to the area of the cells in the inner half of bounding circle”, “lacunarity calculated on the basis of differences in the number of
pixels in each square of the grid for all orientations of the squares, with allowance for the
background pixels taken as zero (only pixels of the image of the object were taken into account in LF), (LE), etc.
The majority of the sentences are too long to be effective and hard to follow.
It could be a good idea to briefly introduce each parameter in this section and then develop each explanation in a new section as results, adding the real images of the cells in which the parameter is applied.

RESULTS
Visual description of cell morphology section: This section describes different type of cellular transformation that cannot be valued without any image that support this description.
Trends in parameter dynamics: when reading this section it is not clear the object of the discussion, the name of each parameter should be mentioned into the text.
Cell types: I would move this information at the beginning of the results.

---

## Round 0.2 · Minor Revisions

Please revise your bibliography to include the suggested references in a new version of the article.

Reviewer 1 ·

Basic reporting

The authors analyzed using a range of linear and non- linear morphological parameters in three stages of cell behavior of immune cells of the marine invertebrate Asterias.
In each stage they described chronological differences in cell complexity, such as size and asymmetry among other. The main statistical found is due to the characterization in several linear and nonlinear cell parameters of the external morphology for all cell types.

Experimental design

This is interesting since of the dynamics of cellular behavior shows a number of transitional forms, which dynamics can also reflect the species-specific features of cellular behavior, the state of the animal's immune status or reaction to the experimental effect.

Validity of the findings

In this context and due to the difficulty of this morphological cellular changes is necessary the use of multiparametric cellular analysis in vitro and in short-term.
The statistical design and correlations are correct supporting the results.

Additional comments

My comment is about the conclusions when a comparative issue es in classical neuronal morphometry which is well developed, in contrast with the fibroblast-like cells, is important to mentioned the morphometrical analysis in microglial cells, in which skeleton was well as fractal analysis is also well studied, I recommend to include these biblio:
Young, K., Morrison, H. Quantifying Microglia Morphology from Photomicrographs of Immunohistochemistry Prepared Tissue Using ImageJ. J. Vis. Exp. (136), e57648, doi:10.3791/57648 (2018).
Morrison Y, et al., Quantitative microglial analyses revels….Scientific Reports | 7: 13211 | DOI:10.1038/s41598-017-13581-z

---

## Round 0.3 · accepted · Accept

Congratulations! Your manuscript is now accepted for publication.